# Effect of Partially Correlated Wind Loading on the Response of Two-Way Asymmetric Systems: The Impact of Torsional Sensitivity and Nonlinear Effects

Adrián López-Ibarra [ID], Adrián Pozos-Estrada *[ID] and Rigoberto Nava-González

Institute of Engineering, Universidad Nacional Autónoma de México, Ciudad de México CP 04510, Mexico; alopezi@iingen.unam.mx (A.L.-I.); rnavag@iingen.unam.mx (R.N.-G.)
* Correspondence: apozose@iingen.unam.mx

**Abstract:** Load eccentricities in structural systems are associated with an increase in the torsional response. Typically, these eccentricities are defined based on the distance between the center of mass and the center of stiffness at a predefined story. If the structural system is subjected to dynamic loading, such as wind loading, instantaneous load eccentricities due to the displacement of the center of mass may occur. An evaluation of this nonlinear effect for two-way asymmetric systems under wind loading is presented in this study. To model the structural systems and the instantaneous load eccentricities, coupled nonlinear differential equations are assembled and solved by using the state space model. The structural systems proposed are subjected to time histories of turbulent wind forces, which are simulated based on a newly developed methodology that includes the correlation of wind forces. The impact of the instantaneous load eccentricities and correlation of wind forces and torsional moment on the wind-induced response of the structural systems analyzed is discussed in detail.

**Keywords:** wind force correlation; wind force simulation; wind-induced torsion; geometric non-linearity; second order effect

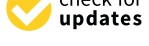



## 1. Introduction

Wind torsional excitation is caused by an imbalance in the instantaneous pressure distributions on all building surfaces and the eccentricity between the elastic and mass centers [1,2]. Several studies have shown that wind-induced torsional motion is frequently associated with external visual cues due to the rotation of the building [3]. The authors of [4] state that the torsional velocity causes relative motion of distant objects that becomes visually apparent and creates the visual sensation of a "swinging horizon" and emphasizes motion perception [5–7]. Other studies on the wind-induced torsional motion of structures have focused on the development of methodologies to evaluate the structural response on structures. For example, Ref. [1] developed a lateral–torsional motion methodology due to wind loads for idealized buildings and emphasized the importance of the torsional response through the fact that the torsional motion was discernable at a much lower level of response than the lateral translational motion. Moreover, Ref. [8] indicated that the mass or the rigidity center may significantly affect the torsional responses at the geometric center of the building, and that the torsional motion contributes significantly to the overall lateral–torsional coupled responses of a tall building with eccentricities in the mass and/or the rigidity center. The authors of [9] carried out an experimental study to evaluate wind-induced torsion in low- and medium-height buildings; their results indicated that applying 75% of the full wind load with an equivalent eccentricity of 15% improved torsion evaluation. More recently, Ref. [10] carried out an experimental study of wind-induced shear, bending, and torsional loads on rectangular tall buildings and emphasized that the torsional moment induced by the wind forces on tall buildings with rectangular cross

sections must be considered during the analysis and structural design stages. It is noted that wind tunnel experiments are of paramount importance in the evaluation of wind-induced loads. In particular, wind tunnel studies have shown to be an excellent tool to evaluate torsional loads on tall buildings with different heights and aspect ratios.

More recently, Ref. [11] studied the torsional response under bidirectional seismic excitations for symmetric and asymmetric linear systems. According to [11], the relative distances between the center of mass (CM) and the center of stiffness (CS) vary with time during the ground motion and the bidirectional seismic excitations lead to torsional responses, and instantaneous load eccentricities occur during the horizontal movement of the CM in the plane that can lead to an extra torsional motion not taken in a count in the seismic codes. The author named this second-order effect the A-Δ effect. It is noted that the evaluation of the A-Δ effect on systems under dynamic wind loading has not been reported in the literature, although several studies have pointed out the importance of the wind-induced torsional response on different types of systems.

Although several studies have provided steps towards an understanding of the importance of the wind-induced response, with emphasis on the torsional response, the evaluation of the impact of the correlation of wind forces and torsional moment, as well as the second-order effect, named the A-Δ effect, still requires further investigation to evaluate its impact on the wind-induced response. The main objective of this study is to evaluate the impact of the correlation of wind forces, torsional moment, and the A-Δ effect on the wind-induced response of two-way asymmetric systems. For the modeling of the structural systems and A-Δ effect, coupled nonlinear differential equations are assembled and solved by using the state space model. The structural systems proposed are subjected to time histories of turbulent wind forces, which are simulated based on a newly developed methodology that includes the correlation of wind forces and the torsional moment. The impact of the correlation of wind forces, torsional moment, and the A-Δ effect on the wind-induced response of the structural systems is discussed in detail.

## 2. Structural Model, Wind-Loading Model, and Analysis Procedure

### 2.1. Equation of Motion and Instantaneous Loads Eccentricities (A-Δ Effect)

For the analysis, actual buildings whose general dimensions are shown in Figure 1a were modeled as three-degree-of-freedom (3DOF) systems with generalized properties; the 3DOF system was composed of a rigid slab supported by frames and walls, as shown in Figure 1b. The structure can experience lateral displacements in the X- and Y-direction (i.e., $u_x$ and $u_y$), as well as rotational displacement, denoted by $u_\theta$.

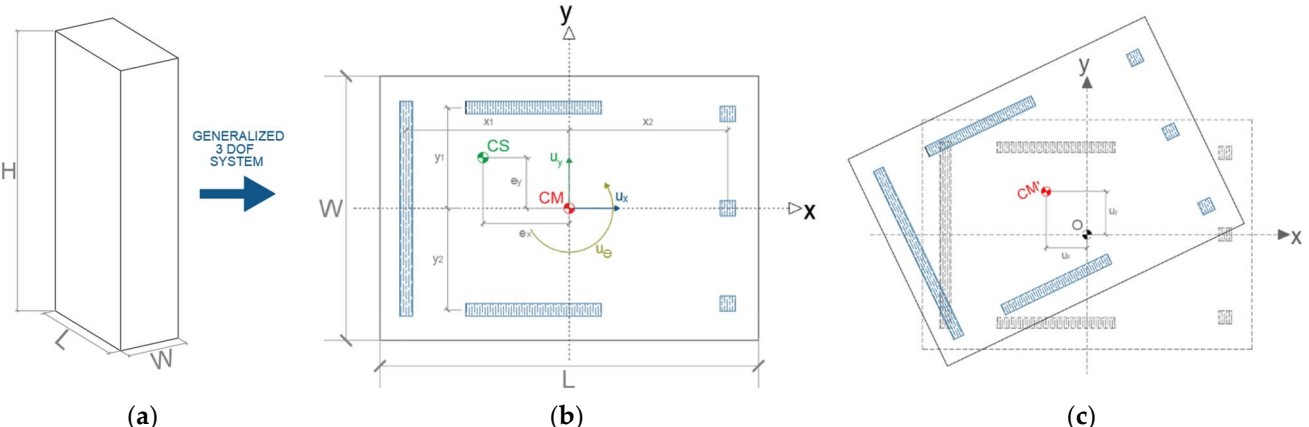

**(a)**　　　　　　　　　　**(b)**　　　　　　　　　　**(c)**

**Figure 1.** Structural systems: (**a**) general dimensions of the buildings; (**b**) plan view of the three-degree-of-freedom system; and (**c**) movement of the CM in time *t* due to wind loads.

The equation of motion without damping of the structure under wind loading is expressed in Equation (1), where *m* is the mass of the structure, *r* is the radius of gyration,

$k_{ij}$ for $I = x, y, \theta$ and $j = x, y, \theta$ represents the elements of the stiffness matrix, $F_x(t)$ and $F_y(t)$ are the wind-induced forces acting in the X- and Y-direction, respectively, and $M(t)$ is the wind-induced torsional moment. The methodology used to simulate the wind-induced forces will be discussed in the next section. A dot on a symbol denotes its temporal derivative (i.e., velocity); and a double-dot on a symbol denotes the second temporal derivative (i.e., acceleration).

$$
\begin{bmatrix} m & 0 & 0 \\ 0 & m & 0 \\ 0 & 0 & mr^2 \end{bmatrix} \begin{Bmatrix} \ddot{u}_x \\ \ddot{u}_y \\ \ddot{u}_\theta \end{Bmatrix} + \begin{bmatrix} k_{xx} & k_{xy} & k_{x\theta} \\ k_{yx} & k_{yy} & k_{y\theta} \\ k_{\theta x} & k_{\theta y} & k_{\theta\theta} \end{bmatrix} \begin{Bmatrix} u_x \\ u_y \\ u_\theta \end{Bmatrix} = \begin{Bmatrix} F_x(t) \\ F_y(t) \\ M(t) \end{Bmatrix} \tag{1}
$$

If Rayleigh damping is considered, Equation (1) can be rewritten as

$$
\begin{Bmatrix} \ddot{u}_x \\ \ddot{u}_y \\ \ddot{u}_\theta \end{Bmatrix} + a_1\omega_x^2 \begin{bmatrix} a_0/(a_1\omega_x^2)+1 & 0 & e_y/r \\ 0 & a_0/(a_1\omega_x^2)+\Omega_y^2 & e_x\Omega_y^2/r \\ e_y/r & e_x\Omega_y^2/r & a_0/(a_1\omega_x^2)+\Omega_\theta^2 \end{bmatrix} \begin{Bmatrix} \dot{u}_x \\ \dot{u}_y \\ \dot{u}_\theta \end{Bmatrix} + \omega_x^2 \begin{bmatrix} 1 & 0 & e_y/r \\ 0 & \Omega_y^2 & \Omega_y^2 e_x/r \\ e_y/r & \Omega_y^2 e_x/r & \Omega_\theta^2 \end{bmatrix} \begin{Bmatrix} u_x \\ u_y \\ u_\theta \end{Bmatrix} = \\ \begin{Bmatrix} F_x(t)/m \\ F_y(t)/m \\ M(t)/mr^2 \end{Bmatrix}, \tag{2}
$$

where $\omega_x = \sqrt{k_{xx}/m}$, $\omega_y = \sqrt{k_{yy}/m}$, and $\omega_\theta = \sqrt{k_{\theta\theta}/mr^2}$ are the circular frequencies associated with each of the degrees of freedom; $\Omega_\theta = \omega_\theta/\omega_x$ and $\Omega_y = \omega_y/\omega_x$ are frequency ratios; $a_0 = 2\zeta\omega_x\omega_y/(\omega_x + \omega_y)$ and $a_1 = 2\zeta/(\omega_x + \omega_y)$ are the coefficients obtained from the Rayleigh damping; and $\zeta$ is the damping ratio.

To incorporate the A-Δ effect into Equation (2), consider Figure 1c that illustrates the movement of the CM with respect to its original position or the CS due to wind loads. The instantaneous movement of CM leads to instantaneous load eccentricities represented by $u_x$ and $u_y$ at time $t$ that can be incorporated into the time-dependent stiffness matrix given by [11]:

$$
K = \begin{bmatrix} k_{xx} & 0 & k_{\theta x} + k_{xx}u_y \\ 0 & k_{yy} & k_{\theta y} - k_{yy}u_x \\ k_{\theta x} + k_{xx}u_y & k_{\theta y} - k_{yy}u_x & k_{\theta\theta} + k_{\theta x}u_y - k_{\theta y}u_x \end{bmatrix}, \tag{3}
$$

If the time-dependent stiffness matrix given in Equation (3) is used in Equation (2), Equation (4) represents coupled nonlinear differential equations that incorporate the A-Δ effect.

$$
\begin{Bmatrix} \ddot{u}_x \\ \ddot{u}_y \\ \ddot{u}_\theta \end{Bmatrix} + a_1\omega_x^2 \begin{bmatrix} a_0/(a_1\omega_x^2)+1 & 0 & e_y/r \\ 0 & a_0/(a_1\omega_x^2)+\Omega_y^2 & e_x\Omega_y^2/r \\ e_y/r & e_x\Omega_y^2/r & a_0/(a_1\omega_x^2)+\Omega_\theta^2 \end{bmatrix} \begin{Bmatrix} \dot{u}_x \\ \dot{u}_y \\ \dot{u}_\theta \end{Bmatrix} + \\ \omega_x^2 \begin{bmatrix} 1 & 0 & (e_y+u_y)/r \\ 0 & \Omega_y^2 & \Omega_y^2(e_x-u_x)/r \\ (e_y+u_y)/r & \Omega_y^2(e_x-u_x)/r & \Omega_\theta^2 + e_y u_y/r - e_x u_x/r \end{bmatrix} \begin{Bmatrix} u_x \\ u_y \\ u_\theta \end{Bmatrix} = \begin{Bmatrix} F_x(t)/m \\ F_y(t)/m \\ M(t)/mr^2 \end{Bmatrix}, \tag{4}
$$

Furthermore, Equation (4) can be rewritten as a set of first-order differential equations in the state-space model, given by

$$
\left\{ \begin{aligned} \dot{y}_1 &= y_4 \\ \dot{y}_2 &= y_5 \\ \dot{y}_3 &= y_6 \\ \dot{y}_4 &= \frac{F_X(t)}{m} - (a_0 + a_1\omega_x^2)y_4 - a_1\omega_x^2 e_y r(ry_6) - \omega_x^2 y_1 - u_x^2 e_y r(ry_3) \\ \dot{y}_5 &= \frac{F_Y(t)}{m} - \left(a_0 + a_1\omega_x^2\Omega_y^2\right)y_5 - a_1\omega_x^2\Omega_y^2 e_x r(ry_6) - \omega_x^2\Omega_y^2 y_2 - \omega_x^2\Omega_y^2 e_y r(ry_3) \\ \dot{y}_6 &= \frac{M(t)}{mr^2} + (-a_1\omega_x^2 e_y r y_4 - a_1\omega_x^2\Omega_y^2 e_x r y_5 - \left(a_0 + a_1\omega_x^2\Omega_\theta^2(ry_6)\right) - \omega_x^2 e_y r y_1 - a_1\omega_x^2\Omega_y^2 e_x r y_2 \\ &\quad - \omega_x^2\Omega_\theta^2(ry_3))/r \end{aligned} \right. , \tag{5}
$$

where $y = \{y_1, y_2, y_3, y_4, y_5, y_6\}^T = \{u_x, u_y, u_\theta, \dot{u}_x, \dot{u}_y, \dot{u}_\theta\}^T$. Equation (5) is solved by using the method proposed by [12].

### 2.2. Wind-Loading Model

For the development of the wind-loading model and to simplify the parametric study, it is considered that the resultant wind forces, denoted by $F_x(t)$, $F_y(t)$, and $M(t)$, acting on the structural model can be modeled by using four uncorrelated wind loads (i.e., $F_1(t)$, $F_2(t)$, $F_3(t)$, and $F_4(t)$) that are acting at different points of the structural model, as shown in Figure 2. The power spectral density (PSD) functions of the uncorrelated wind loads are given by Equation (6):

$$S_n(f) = \begin{cases} Af^{-(1+\alpha)} & f_L \leq f \leq f_u \\ 0 & \text{otherwise} \end{cases}, \tag{6}$$

where $f$ is frequency in Hz, $f_L$ and $f_u$ are upper and lower bounds of $f$, $\alpha$ is an exponent that depends on the direction of the wind velocity, and $A$ is a normalization constant such that the integration of the PSD function equals one. This PSD function has been previously used in the studies [13,14], since boundary layer wind tunnel test results indicate that such a PSD function provides sufficiently accurate characterization of the fluctuating wind force in the inertial subrange, which is of interest in the present study.

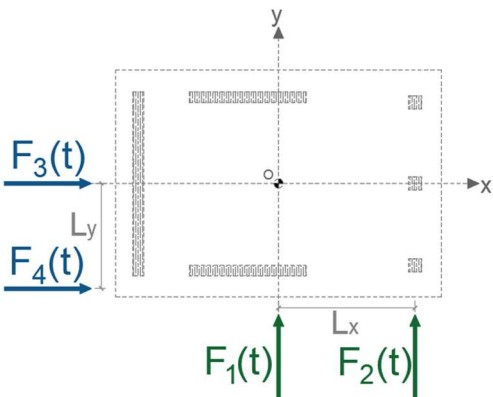

**Figure 2.** Representation of the uncorrelated wind forces over the structural system.

Based on the uncorrelated wind forces shown in Figure 2, the resultant forces $F_x(t)$ and $F_y(t)$ and the torsional moment with respect to the origin $M(t)$ are given by

$$F_x(t) = F_3(t) + F_4(t), \tag{7a}$$

$$F_y(t) = F_1(t) + F_2(t), \tag{7b}$$

$$M(t) = F_2(t) \cdot L_x + F_4(t) \cdot L_y, \tag{7c}$$

where $L_x$ and $L_y$ are distances between the lateral forces $F_1$ and $F_2$, and $F_3$ and $F_4$, respectively.

Several studies have shown that the resultant wind forces (i.e., $F_x(t)$, $F_y(t)$, and $M(t)$) present a certain degree of correlation. For example, Refs. [15–17], based on wind tunnel experiments, showed that there is low correlation between the maximum values of the along-wind force and the across-wind force, high correlation between the maximum values of the along-wind force and the torsional moment, and very low correlation between the maximum values of the across-wind force and the torsional moment. If the correlation

between $F_x(t)$ and $M(t)$, $\rho_{xM}$, and between $F_y(t)$ and $M(t)$, $\rho_{yM}$, is of interest, Equation (7) can be rewritten as

$$F_x(t) = c_1 \cdot \sqrt{(1 - \rho_{xM}{}^2) - [\rho_{yM} \cdot \rho_{xM} \cdot r_\sigma] \cdot r_e} \cdot F_3(t) + c_1 \cdot \sqrt{\rho_{xM}{}^2 + [\rho_{yM} \cdot \rho_{xM} \cdot r_\sigma] \cdot r_e} \cdot F_4(t), \quad (8a)$$

$$F_y(t) = c_2 \cdot \sqrt{(1 - \rho_{yM}{}^2) - \left[\rho_{yM} \cdot \rho_{xM} \cdot \frac{1}{r_\sigma}\right] \cdot \frac{1}{r_e}} \cdot F_1(t) + c_2 \cdot \sqrt{\rho_{yM}{}^2 + \left[\rho_{yM} \cdot \rho_{xM} \cdot \frac{1}{r_\sigma}\right] \cdot \frac{1}{r_e}} \cdot F_2(t), \quad (8b)$$

$$M(t) = c_1 \cdot \sqrt{\rho_{xM}{}^2 + [\rho_{yM} \cdot \rho_{xM} \cdot r_\sigma] \cdot r_e} \cdot F_4(t) \cdot L_y + c_2 \cdot \sqrt{\rho_{yM}{}^2 + \left[\rho_{yM} \cdot \rho_{xM} \cdot \frac{1}{r_\sigma}\right] \cdot \frac{1}{r_e}} \cdot F_2(t) \cdot L_x, \quad (8c)$$

where $c_1$ and $c_2$ are model parameters; $r_\sigma$ is the ratio of the parameters $c_1$ and $c_2$; and $r_e$ is the ratio between the lengths $L_x$ and $L_y$. Details about the development of Equation (8) are given in Appendix A.

To evaluate $F_x(t)$, $F_y(t)$, and $M(t)$, wind forces $F_i(t)$ ($i = 1, \ldots, 4$) are first simulated by using the spectral representation method (SRM) [18]. It is noted that more sophisticated simulation methods can also be used [19]; however, for simplicity, the characterization of the wind forces adopted in this study followed a practical approach. To apply the SRM, given the PSD function of the wind force (Equation (6)), samples of $F_i(t)$ ($i = 1, \ldots, 4$) can be obtained as

$$F_i(t) = \sqrt{2} \sum_{j=1}^{N} \sqrt{S_n(f_j) \Delta f} \sin(2\pi f_j t + \phi_j), \quad (9)$$

where $f_j = j \times \Delta f$, $j = 1, 2, \ldots, N$, $\Delta f = 1/(N\Delta t)$, $\Delta t$ is the time increment, and $\phi_j$ is a uniformly distributed random variable within 0 to $2\pi$.

The evaluated samples of $F_x(t)$, $F_y(t)$, and $M(t)$ can then be used in Equation (5) to calculate the time history response of the structure with and without the A-$\Delta$ effect.

### 2.3. Analysis Procedure

The influence of the instantaneous load eccentricities on the response of the structural models evaluated in this study can be measured by employing the response ratios of the maximum peak responses of the structure with the instantaneous load eccentricities (A-$\Delta$ effect), and the maximum peak responses without the influence of the instantaneous load eccentricities. Further, it is noted that the excessive wind-induced response could lead to various problems associated with human comfort [6,7,20]; for this reason, Equation (10) defines the response ratios in terms of displacement and acceleration.

$$R_X = \max_{t \in |0, T_r|} |u_{xAD}(t)| / \max_{t \in |0, T_r|} |u_x(t)|, \quad (10a)$$

$$R_Y = \max_{t \in |0, T_r|} |u_{yAD}(t)| / \max_{t \in |0, T_r|} |u_y(t)|, \quad (10b)$$

$$R_{\ddot{X}} = \max_{t \in |0, T_r|} \left|\ddot{u}_{xAD}(t) \mp \ddot{\theta}_{AD}(t) r\right| / \max_{t \in |0, T_r|} \left|\ddot{u}_x(t) \mp \ddot{\theta}(t) r\right|, \quad (10c)$$

$$R_{\ddot{Y}} = \max_{t \in |0, T_r|} \left|\ddot{u}_{yAD}(t) \mp \ddot{\theta}_{AD}(t) r\right| / \max_{t \in |0, T_r|} \left|\ddot{u}_y(t) \mp \ddot{\theta}(t) r\right|, \quad (10d)$$

where $T_r$ denotes the duration of the numerical analysis (10 min in this study). In Equation (10), symbols with the additional subscript $AD$ indicate that the structural response includes the A-$\Delta$ effect. For a given structure and predefined parameters to characterize the wind-loading model, the analysis procedure for assessing the influence of the instantaneous load eccentricities and correlation of wind forces and torsional moment on the response can be summarized as follows:

(1)  Simulate $F_x(t)$, $F_y(t)$ and $M(t)$ by using Equation (8);
(2)  Solve Equation (5) for the structure without and with the A-$\Delta$ effect to obtain the time history of the responses using Gear's method [12];
(3)  Evaluate the response ratios (i.e., $R_X$, $R_Y$, $R_{\ddot{X}}$, and $R_{\ddot{Y}}$) defined in Equation (10);
(4)  Repeat steps 1 to 3 for all the structures considered.

Furthermore, the previous procedure can also be used to calculate the statistics of the response with and without the A-$\Delta$ effect.

## 3. Parametric Analysis and Results

### 3.1. Structural Characteristics and Peak Responses

The analyses were carried out with six structural models, which represent actual structures approximated by 3DOF systems whose characteristics are defined by the generalized properties of actual tall buildings. The dynamic and geometric characteristics of the structural systems are summarized in Table 1. The parameters $L_x$ and $L_y$ employed in the wind-loading model are also included in Table 1.

**Table 1.** Parameters of the structural and wind model.

| Model | Mass (kg) | $\omega_x$ (rad/s) | $\omega_y$ (rad/s) | $H$ (m) | $L$ (m) | $W$ (m) | $H/L$ | $H/W$ | $L_x$ (m) | $L_y$ (m) | $r_e = L_x/L_y$ |
|---|---|---|---|---|---|---|---|---|---|---|---|
| 1 | 2,281,509 | 0.925 | 0.945 | 170 | 40 | 33 | 4.3 | 5.2 | 12 | 11 | 1.09 |
| 2 | 3,516,347 | 1.259 | 0.788 | 262 | 55 | 25 | 4.8 | 10.5 | 35 | 30 | 1.17 |
| 3 | 2,180,727 | 1.179 | 1.033 | 140 | 38 | 34 | 3.7 | 4.1 | 15 | 13 | 1.15 |
| 4 | 827,663 | 0.980 | 1.045 | 116 | 36 | 19 | 3.2 | 6.1 | 28 | 25 | 1.12 |
| 5 | 479,315 | 1.098 | 1.559 | 101 | 29 | 14 | 3.5 | 7.2 | 20 | 18 | 1.11 |
| 6 | 2,519,208 | 0.753 | 0.753 | 200 | 30 | 30 | 6.7 | 6.7 | 20 | 20 | 1.00 |

For all the numerical analyses, the model parameters $c_1$ and $c_2$ were selected such that the mean peak acceleration was within acceleration thresholds of perception [3,7].

For the parametric analysis, the natural frequency in torsion ($\omega_\theta$) varied for each model as a function of the uncoupled torsional ratio ($\Omega_\theta = \omega_\theta / \omega_x$), which varied from 0.8 to 2.0 with increments equal to 0.1. The damping ratios for the sway modes ($\zeta_x$ and $\zeta_y$) and torsional mode ($\zeta_\theta$) were set equal to 1%.

To assess the influence of the correlation between $F_x(t)$ and $M(t)$, and $F_y(t)$ and $M(t)$ (i.e., $\rho_{FxM}$ and $\rho_{FyM}$) on the mean peak responses, for the moment consider that the structural model 1 of Table 1 is subjected to wind loads calculated with Equation (8) for five different correlation pairs [i.e., (0, 1), (0, 0.25), (0.25, 0), (0.5, 0.5), and (0.75, 0.5)] and the structural response is calculated by solving Equation (5). To account for the uncertainty in the wind-induced response, the analyses were repeated 30 times to evaluate the mean peak responses. To show the impact of $\rho_{FxM}$ and $\rho_{FyM}$ on the mean peak responses, Figure 3 presents the variation of the mean peak displacement, rotation, and acceleration of the structural model 1 without the A-$\Delta$ effect for $\Omega_\theta = 1$.

It can be observed in Figure 3 that the mean peak responses shown are affected by $\rho_{FxM}$ and $\rho_{FyM}$. The analysis carried out for the structural model 1 was repeated, but considering the rest of the structural models shown in Table 1. Figure 4 presents a comparison of the mean peak displacement, rotation, and acceleration for all the models presented in Table 1 for $\Omega_\theta = 1$.

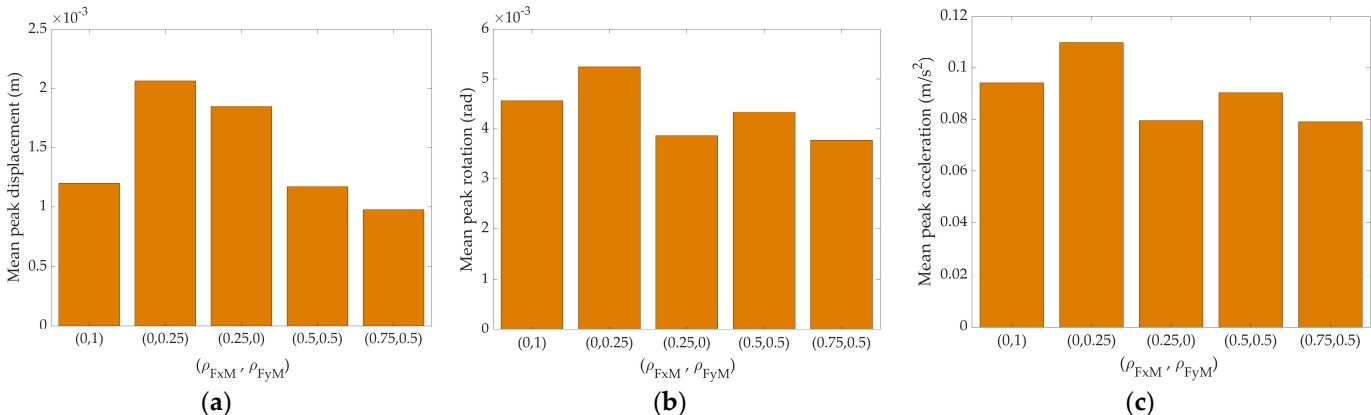

**Figure 3.** Mean peak responses of the structural models without the A−Δ effect: (**a**) displacement; (**b**) rotation; (**c**) acceleration.

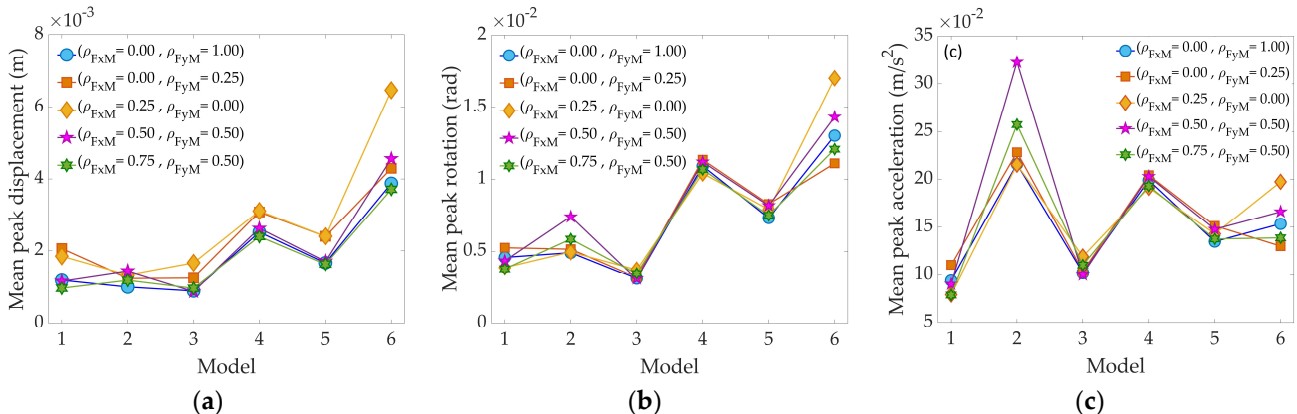

**Figure 4.** Comparison of mean peak responses of the structural models considering different correlation pairs without the A−Δ effect: (**a**) displacement; (**b**) rotation; (**c**) acceleration.

It can be observed in Figure 4 that the consideration of $\rho_{FxM}$ and $\rho_{FyM}$ affects the mean peak response. It is worth noting that the dimensions of the buildings modelled as 3DOF systems with generalized properties have important implications in the wind-induced response. As shown in Table 1, the heights of the buildings range from 101 to 262 m, and the aspect ratios (i.e., $H/L$ and $H/W$) are within 3.2 to 10.5, which indicate that the structural models are very sensitive to the dynamic effects of the wind loading. As indicated in [10], the aspect ratios ($H/L$ and $H/W$) are of paramount importance in the evaluation of the wind-induced response with particular attention to the torsional moment.

### 3.1.1. Impact of $\Omega_\theta$ and ($\rho_{FxM}$, $\rho_{FyM}$) on the Mean Peak Responses

To further evaluate the influence of $\Omega_\theta$ and the correlation pair ($\rho_{FxM}$, $\rho_{FyM}$) on the structural response, the analysis procedure described in Section 3.1 was repeated with the $\Omega_\theta$ values varying from 0.8 to 2.

Figure 5 presents the variation of the mean peak displacement, rotation, and acceleration with respect to $\Omega_\theta$ and ($\rho_{FxM}$, $\rho_{FyM}$). The plots of the mean peak displacement shown in Figure 5 indicate that $\Omega_\theta$ and ($\rho_{FxM}$, $\rho_{FyM}$) have an important impact, and that in some cases the combination of certain values of $\Omega_\theta$ and ($\rho_{FxM}$, $\rho_{FyM}$) produces a greater mean peak displacement for all the structural models analyzed. It can also be observed in Figure 5 that, as expected, the most important parameter in the calculation of the mean peak rotation is $\Omega_\theta$, and that only in the cases where $\Omega_\theta < 1$, the importance of considering ($\rho_{FxM}$, $\rho_{FyM}$) gain relevance. The plots of mean peak acceleration shown in Figure 5 are relatively uniform along $\Omega_\theta$ and ($\rho_{FxM}$, $\rho_{FyM}$), as this was predefined in the scaling of the

wind forces. The results presented in Figure 5 are concordant with previous experimental studies carried out by [15–17], where the correlation of wind forces were shown to have a very important impact on the wind-induced response.

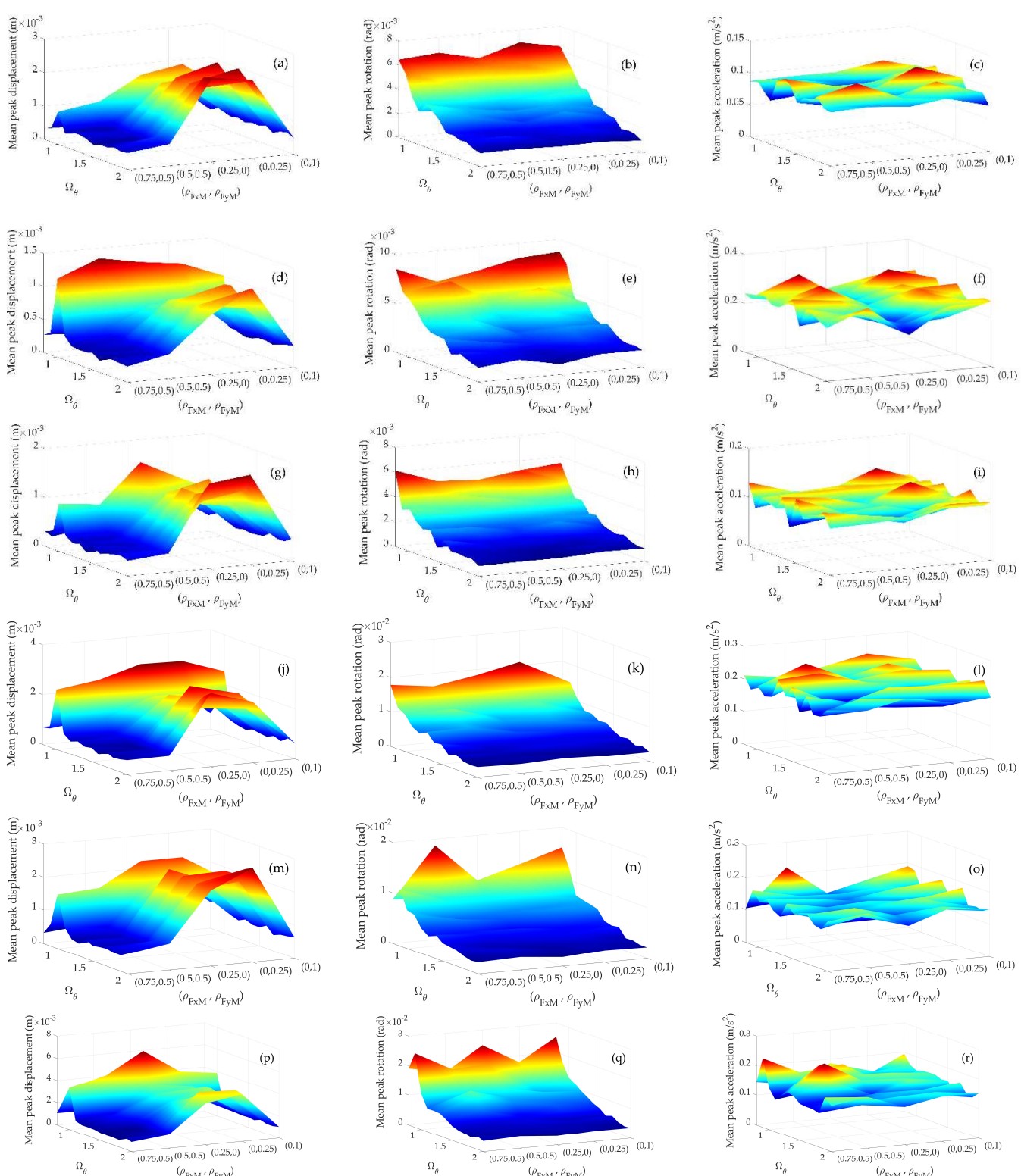

**Figure 5.** Comparison of mean peak responses of the structural models considering different correlation pairs without the A−Δ effect: (**a**,**d**,**g**,**j**,**m**) and (**p**) displacement; (**b**,**e**,**h**,**k**,**n**) and (**q**) rotation; (**c**,**f**,**i**,**l**,**o**) and (**r**) acceleration.

### 3.1.2. Impact of the A-Δ effect, $\Omega_\theta$, and $(\rho_{FxM}, \rho_{FyM})$ on the Peak Responses

To evaluate the influence of the nonlinear A-Δ effect, the nonlinear response was calculated with Equation (5) in order to calculate the response ratios from Equation (10).

Figure 6 presents the variation of $R_X$ and $R_{\ddot{X}}$. In Figure 6, the boxplots present statistics of the 30 samples of $R_X$ and $R_{\ddot{X}}$ for different values of $\Omega_\theta$ and $(\rho_{FxM}, \rho_{FyM})$ for all the structural models considered. For each boxplot, the median of the samples is indicated with a central mark, the lower and upper bounds of the box indicate the 25th and 75th percentiles, the most extreme data points not considered outliers are represented by whiskers, and the outliers are plotted individually using the cross symbol [21]. In Figure 6a, a rectangle is included to group the first five boxplots, which represent the variation of $R_X$ for different values of $(\rho_{FxM}, \rho_{FyM})$ for $\Omega_\theta$ equal to 0.8. The subsequent boxplots were used to show the impact of $\Omega_\theta$ on $R_X$ for different levels of correlation of the wind forces.

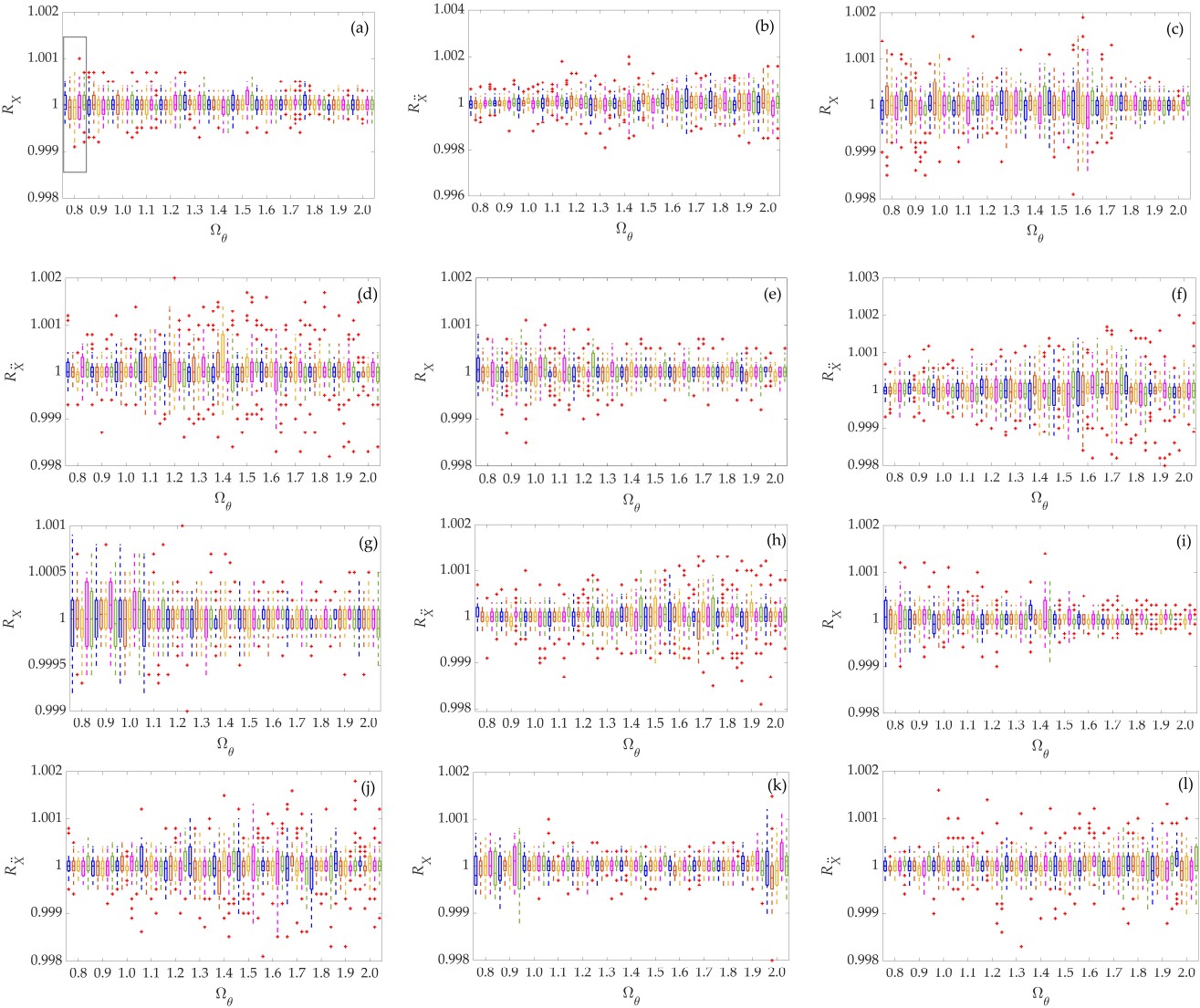

**Figure 6.** Influence of the A−Δ effect on the displacement and acceleration in the X-direction: (**a**,**b**) Model 1; (**c**,**d**) Model 2; (**e**,**f**) Model 3; (**g**,**h**) Model 4; (**i**,**j**) Model 5; (**k**,**l**) Model 6. The correlation of the wind forces are indicated by the color of the boxplot: blue [$\rho_{FxM} = 0$, $\rho_{FyM} = 1$], red [$\rho_{FxM} = 0$, $\rho_{FyM} = 0.25$], yellow [$\rho_{FxM} = 0.25$, $\rho_{FyM} = 0$], cyan [$\rho_{FxM} = 0.5$, $\rho_{FyM} = 0.5$], green [$\rho_{FxM} = 0.75$, $\rho_{FyM} = 0.5$].

It can be observed in Figure 6 that, for all the structural models analyzed, the median of the $R_X$ values is very close to one for all the values considered of $\Omega_\theta$. It is also observed

that even the extreme data, as well as the outliers, oscillate around one. The latter indicates that the A-Δ effect does not have an important impact on the wind-induced displacement of the models considered. Similar to the results presented in Figure 5, the results presented in Figure 6 show that the correlation coefficient between the wind forces (i.e., $\rho_{FxM}$ and $\rho_{FyM}$) has the most important impact on the $R_X$ ratios. Similar observations are drawn for the $R_{\ddot{X}}$ ratios, except that greater scatter is observed than those for the $R_X$ values; however, the variation of $R_{\ddot{X}}$ does not have an important impact on the wind-induced acceleration, indicating that the A-Δ effect is negligible.

Similar results to those presented in Figure 6 were obtained for the *Y*-direction (i.e., $R_Y$ and $R_{\ddot{Y}}$), and for this reason they are not included.

It is noted that the $R_{\ddot{X}}$ values shown in Figure 6 include the effect of the wind-induced angular acceleration ($\ddot{\theta}(t)$), which is related to rotational velocity ($\dot{\theta}(t)$). Since $\dot{\theta}(t)$ could be of concern for certain cases [14], to further evaluate the impact of the A-Δ effect on $\dot{\theta}(t)$, Figure 7 presents a comparison of $\dot{\theta}(t)$ without and with the A-Δ effect for all the structural models considered for selected values of $\Omega_\theta$ and ($\rho_{FxM}, \rho_{FyM}$). It can be observed in Figure 7 that the rotational velocity with and without the A-Δ effect is practically the same, indicating that the A-Δ effect has a marginal impact on the wind-induced rotational velocity. Similar observations were drawn for different values of $\Omega_\theta$ and ($\rho_{FxM}, \rho_{FyM}$), and for this reason they are not presented.

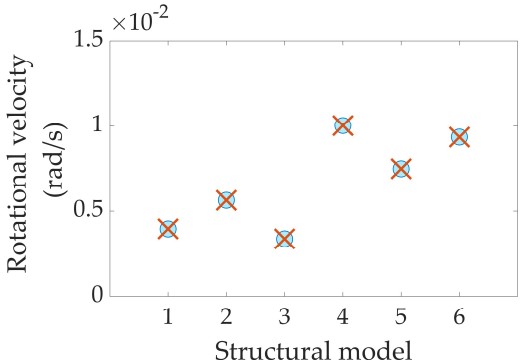

**Figure 7.** Comparison of rotational velocity without and with the A−Δ effect for all the structural models considered for $\Omega_\theta = 1$, and $\rho_{FxM} = 0.75$ and $\rho_{FyM} = 0.5$. Blue dots: without the A−Δ effect; cross: with the A−Δ effect.

## 4. Conclusions

Numerical analyses were conducted to investigate whether the nonlinear effect, or A-Δ effect, and the correlation of the wind forces and the torsional moment had an impact on the wind-induced response. For the analysis, a newly developed wind-loading model, which includes the effect of correlation between the wind forces (forces and moments) was used to simulate wind forces that were applied to generalized structural models with and without the A-Δ effect. The numerical results indicate that the impact of the A-Δ effect on the wind-induced response is negligible and that the most important effect on the wind-induced response is due to the correlation between the wind forces and the torsional moment. More specifically, it is concluded that:

- In all the structural models analyzed, the median and the extreme data, as well as the outliers of the response ratio ($R_X$, $R_{\ddot{X}}$, $R_Y$, and $R_{\ddot{Y}}$) values are very close to one for all the values considered of $\Omega_\theta$. The latter indicates that the A-Δ effect does not have an important impact on wind-induced displacement and acceleration.
- The correlation coefficient between the wind forces (i.e., $\rho_{FxM}$ and $\rho_{FyM}$) has the most important impact on the response, indicating that different levels of correlation of the wind forces and the torsional moment have to be considered in the evaluation of the wind-induced response.

- The wind-induced rotational velocity with and without the A-Δ effect is practically the same, indicating that the A-Δ effect has a marginal impact on it.
- The results presented in this study show that the A-Δ effect has a marginal impact on the wind-induced response; however, the evaluation of the wind-induced response considering the nonlinear behavior of the structure (i.e., material or element nonlinearity) is very scarce in the relevant literature. Different to the nonlinear response of structures under seismic action, where the nonlinear response is likely to reduce compared to the linear response, wind-induced response will not necessarily behave in the same way, as nonlinearity may cause the structure to become more flexible with a consequent increase in the dynamic response of the structure. Further analyses are required to evaluate the impact of the material and element nonlinearity on tall buildings subjected to partially correlated wind loads.

**Author Contributions:** Conceptualization, A.L.-I. and A.P.-E.; Methodology, A.L.-I. and A.P.-E.; Investigation, A.L.-I., A.P.-E., and R.N.-G.; Writing—original draft, A.L.-I., A.P.-E., and R.N.-G.; Writing—review and editing, A.P.-E. and R.N.-G.; Resources, A.L.-I. and A.P.-E.; Supervision, A.L.-I. and A.P.-E. All authors have read and agreed to the published version of the manuscript.

**Funding:** This work was supported by UNAM-PAPIIT IN103422.

**Data Availability Statement:** No data are available for this study.

**Acknowledgments:** The financial support of the National Council for Science and Technology (CONACYT) of Mexico and the Institute of Engineering are gratefully acknowledged.

**Conflicts of Interest:** The authors declare no conflict of interest.

## Appendix A

The development of the wind-loading model is presented below.

Consider Figure 2. The summation of forces along the X- and Y-direction and the summation of moments about the origin (O) yields:

$$F_x(t) = F_3(t) + F_4(t) \tag{A1}$$

$$F_y(t) = F_1(t) + F_2(t) \tag{A2}$$

$$M(t) = F_2(t) \cdot L_x + F_4(t) \cdot L_y \tag{A3}$$

For the rest of the derivation of the wind-loading model and without loss of generality, the reference to time, *t*, has been dropped. The variances of the wind forces $F_x$, $F_y$, and $M$ are given by:

$$\sigma_{F_x}^2 = E\left[F_3^2\right] + E\left[F_4^2\right] = \sigma_{F_3}^2 + \sigma_{F_4}^2 \tag{A4}$$

$$\sigma_{F_y}^2 = E\left[F_1^2\right] + E\left[F_2^2\right] = \sigma_{F_1}^2 + \sigma_{F_2}^2 \tag{A5}$$

$$\sigma_M^2 = E\left[F_4^2 \cdot L_y^2\right] + E\left[F_2^2 \cdot L_x^2\right] = \sigma_{F_2}^2 \cdot L_x^2 + \sigma_{F_4}^2 \cdot L_y^2 \tag{A6}$$

where E[·] is the expected operator; and $\sigma_{F_1}^2$, $\sigma_{F_2}^2$, $\sigma_{F_3}^2$, and $\sigma_{F_4}^2$ are the variances of the uncorrelated wind forces $F_1$, $F_2$, $F_3$, and $F_4$. The correlation coefficient between $F_x$ and $M$ ($\rho_{xM}$), and $F_y$ and $M$ ($\rho_{yM}$), are given, respectively, by:

$$\rho_{xM} = \frac{E\left[F_4^2\right] \cdot L_y}{\sigma_{F_x} \cdot \sigma_M}, \ \rho_{yM} = \frac{E\left[F_2^2\right] \cdot L_x}{\sigma_{F_y} \cdot \sigma_M}$$

From $\rho_{xM}$ and $\rho_{yM}$, $\sigma_{F_4}^2$ and $\sigma_{F_2}^2$ are obtained, respectively, as:

$$\sigma_{F_4}^2 = E\left[F_4^2\right] = \rho_{xM} \cdot \frac{\sigma_{F_x} \cdot \sigma_M}{L_y}, \quad \sigma_{F_2}^2 = E\left[F_2^2\right] = \rho_{yM} \cdot \frac{\sigma_{F_y} \cdot \sigma_M}{L_x}$$

Substituting $\sigma_{F_4}^2$ and $\sigma_{F_2}^2$ into Equations (A1)–(A3) yields:

$$\sigma_{F_3}^2 = \sigma_{F_x}^2 - \rho_{xM} \cdot \frac{\sigma_{F_x} \cdot \sigma_M}{L_y}$$
$$\sigma_{F_1}^2 = \sigma_{F_y}^2 - \rho_{yM} \cdot \frac{\sigma_{F_y} \cdot \sigma_M}{L_x}$$
$$\sigma_M = \rho_{yM} \cdot \sigma_{F_y} \cdot L_x + \rho_{xM} \cdot \sigma_{F_x} \cdot L_y$$

It is noted that $F_1$, $F_2$, $F_3$, and $F_4$ are uncorrelated wind forces with unit variance and zero mean. To scale such wind forces to predefined values, $F_1$, $F_2$, $F_3$, and $F_4$ are scaled by multiplying for the corresponding standard deviation. Equations (A1)–(A3) are rearranged to include the scaling factors (i.e., standard deviation), and the new equations are written as:

$$F_x = \sigma_{F_3} \cdot F_3 + \sigma_{F_4} \cdot F_4 \tag{A7}$$

$$F_y = \sigma_{F_1} \cdot F_1 + \sigma_{F_2} \cdot F_2 \tag{A8}$$

$$M = \sigma_{F_2} \cdot F_2 \cdot L_x + \sigma_{F_4} \cdot F_4 \cdot L_y \tag{A9}$$

Substituting $\sigma_{F_1}$, $\sigma_{F_2}$, $\sigma_{F_3}$, and $\sigma_{F_4}$ into Equations (A7)–(A9) yields:

$$F_x = \sqrt{\sigma_{F_x}^2 - \rho_{xM} \cdot \frac{\sigma_{F_x} \cdot \sigma_M}{L_y}} \cdot F_3 + \sqrt{\rho_{xM} \cdot \frac{\sigma_{F_x} \cdot \sigma_M}{L_y}} \cdot F_4. \tag{A10}$$

$$F_y = \sqrt{\sigma_{F_y}^2 - \rho_{yM} \cdot \frac{\sigma_{F_y} \cdot \sigma_M}{L_x}} \cdot F_1 + \sqrt{\rho_{yM} \cdot \frac{\sigma_{F_y} \cdot \sigma_M}{L_x}} \cdot F_2. \tag{A11}$$

$$M = \sqrt{\rho_{xM} \cdot \frac{\sigma_{F_x} \cdot \sigma_M}{L_y}} \cdot F_4 \cdot L_y + \sqrt{\rho_{yM} \cdot \frac{\sigma_{F_y} \cdot \sigma_M}{L_x}} \cdot F_2 \cdot L_x. \tag{A12}$$

By using $\sigma_M$ and the following definitions $\sigma_{F_x} = c_1$; $\sigma_{F_y} = c_2$; $\frac{\sigma_{F_y}}{\sigma_{F_x}} = r_\sigma$; $\frac{\sigma_{F_x}}{\sigma_{F_y}} = \frac{1}{r_\sigma}$; $\frac{L_x}{L_y} = r_e$; $\frac{L_y}{L_x} = \frac{1}{r_e}$, Equations (A10)–(A12) can be rewritten as:

$$F_x = c_1 \cdot \sqrt{(1 - \rho_{xM}{}^2) - \left[\rho_{yM} \cdot \rho_{xM} \cdot r_\sigma\right] \cdot r_e} \cdot F_3 + c_1 \cdot \sqrt{\rho_{xM}{}^2 + \left[\rho_{yM} \cdot \rho_{xM} \cdot r_\sigma\right] \cdot r_e} \cdot F_4. \tag{A13}$$

$$F_y = c_2 \cdot \sqrt{(1 - \rho_{yM}{}^2) - \left[\rho_{yM} \cdot \rho_{xM} \cdot \frac{1}{r_\sigma}\right] \cdot \frac{1}{r_e}} \cdot F_1 + c_2 \cdot \sqrt{\rho_{yM}{}^2 + \left[\rho_{yM} \cdot \rho_{xM} \cdot \frac{1}{r_\sigma}\right] \cdot \frac{1}{r_e}} \cdot F_2 \tag{A14}$$

$$M = c_1 \cdot \sqrt{\rho_{xM}{}^2 + \left[\rho_{yM} \cdot \rho_{xM} \cdot r_\sigma\right] \cdot r_e} \cdot F_4 \cdot L_y + c_2 \cdot \sqrt{\rho_{yM}{}^2 + \left[\rho_{yM} \cdot \rho_{xM} \cdot \frac{1}{r_\sigma}\right] \cdot \frac{1}{r_e}} \cdot F_2 \cdot L_x. \tag{A15}$$

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
