# Peer review of "Effect of Partially Correlated Wind Loading on the Response of Two-Way Asymmetric Systems: The Impact of Torsional Sensitivity and Nonlinear Effects"

_applsci, doi:10.3390/app13116421_

Round 1

Reviewer 1 Report

An interesting article suitable for this magazine.

Has been an experiment done? If so, in which wind tunnel and what were the parameters of the modeled structure.

Table 1 - dimensions from figures no. 1 and 2. are not defined precisely

In the final discussion, I expect a larger discussion on the dimension of the height of the structure in relation to the floor plan dimensions of the structure.

Author Response

Replies to Reviewer #1

An interesting article suitable for this magazine.

We thank Reviewer #1 for reading and commenting on our submitted manuscript. The constructive comments and suggestions are appreciated. The constructive comments have allowed us to provide an improved manuscript. Our replies and how we revised our manuscript are detailed in the following.

Q1. Has been an experiment done? If so, in which wind tunnel and what were the parameters of the modeled structure.

R1. We thank the reviewer for this interesting question. For this particular study, we did not carry out experimental analysis; however, we would like to share with the reviewer that some authors of the present study have participated in wind tunnel experiments to study the shear, bending and torsional wind induced loads on tall buildings. Such wind tunnel experiments were carried out in the boundary layer wind tunnel (BLWT) laboratory located at the main campus of the National Autonomous University of Mexico. The modeled structures were rectangular buildings with different heights (from 300 mm to 500 mm with increments of 50 mm), the geometric scale selected was 1/400 for all models. More information about the experimental tests can be found in Guzman-Solis et al. (2020) cited in lines 42-45.

To emphasize the importance of wind tunnel experiments to evaluate wind-induced loads, with particular emphasize in torsional loads, the following paragraph was included in the revised version of the manuscript.

Lines 45 - 48 of the revised manuscript.

‘It is noted that wind tunnel experiments are of paramount importance in the evaluation of wind-induced loads. In particular, wind tunnel studies have shown to be an excellent tool to evaluate torsional loads on tall buildings with different heights and aspect ratios.’

Q2. Table 1 - dimensions from figures no. 1 and 2. are not defined precisely

R2. Agreed. We have improved the readability of Figures 1 and 2 to clarify the dimensions presented in Table 1. We have also updated the information presented in Table 1.

Q3. In the final discussion, I expect a larger discussion on the dimension of the height of the structure in relation to the floor plan dimensions of the structure.

R3. Agreed. We have included the following paragraph in the revised version of the manuscript.

Lines 215 - 222 of the revised manuscript.

‘The dimensions of the buildings modelled as 3DOF systems with generalized properties have important implications in the wind-induced response. As shown in Table 1, the heights of the buildings range from 101 to 262 m, and the aspect ratios (i.e., H/L and H/W) are within 3.2 to 10.5, which indicate that the structural models are very sensitive to the dynamic effects of the wind loading.  As indicated in Guzman-Solis et al. (2020), the aspect ratios (H/L and H/W) are of paramount importance in the evaluation of the wind-induced response with particular attention to the torsional moment. The dynamic properties and the geometric characteristics of the structural models analyzed together with different levels of correlation of the wind forces causes important variations of the mean peak responses.’ 

Reviewer 2 Report

The manuscript investigates the effect of eccentricity on the response of structures to the torsional wind loading through time history analysis. The research is significant and the manuscript is well written. The authors may consider the following comments:

1) The numerical analyses are usually validated/verified by experimental analyses. Verification can also be performed by previous numerical studies in the literature. Please consider addressing this issue in the paper.

2) How are the results affected if the nonlinear behavior of structures are considered in modeling? i.e. material or element nonlinearity. Please discuss in the conclusion section.

Author Response

Replies to Reviewer #2

The manuscript investigates the effect of eccentricity on the response of structures to the torsional wind loading through time history analysis. The research is significant and the manuscript is well written. The authors may consider the following comments:

We thank Reviewer #2 for reading and commenting on our manuscript. The constructive comments have allowed us to provide an improved manuscript. Our replies and how we revised our manuscript are detailed in the following.

Q1) The numerical analyses are usually validated/verified by experimental analyses. Verification can also be performed by previous numerical studies in the literature. Please consider addressing this issue in the paper.

R1. We thank the reviewer for this remark, which we find very valuable. For this particular study, we did not carry out experimental analyses; however, we would like to share with the reviewer that some authors of the present study have participated in wind tunnel experiments to study the shear, bending and torsional wind induced loads on tall buildings (Guzman-Solis et al., 2020). Some of the findings of such wind tunnel studies together with those reported by Tamura et al. (2003, 2008, 2014) show that correlation of wind forces has a very important role on the wind-induced response. The numerical results presented in the present study are concordant with the previous observations. With regard to the verification of the A-D effect, we have only found one study that discusses with detail such second order effect (Hong, 2013); however, the action considered in that study was of the seismic type and for that reason a direct comparison with the results of the present study would be inadequate. Indeed, the lack of studies to evaluate the impact of the correlation of wind forces and torsional moment as well as the A-Δ effect motivated us to carry out the present study.

To emphasize the importance of validating numerical analyses by means of experimental studies or by previous numerical studies in the literature, we have included the following paragraph in the revised version of the manuscript.

Lines 240 - 242 of the revised manuscript.

‘The results presented in Figure 5 are concordant with previous experimental studies carried out by Tamura et al. (2003, 2008, 2014) where the correlation of wind forces showed to have a very important impact on the wind-induced response.’

Q2) How are the results affected if the nonlinear behavior of structures are considered in modeling? i.e. material or element nonlinearity. Please discuss in the conclusion section.

R2. Agreed. This is a very relevant question as the nonlinear behavior of structures has gained much attention in performance-based wind design. We have included the following lines in the Conclusions section.

Lines 310 - 320 of the revised manuscript.

 ‘The results presented in this study showed that the A-D effect has a marginal impact on the wind-induced response; however, the evaluation of the wind-induced response considering the nonlinear behavior of the structure (i.e., material or element nonlinearity) is very scarce in the relevant literature. Different to the nonlinear response of structures under seismic action, where the nonlinear response is likely to reduce as compare to the linear response, wind-induced response will not necessarily behave in the same way, as nonlinearity may cause the structure to become more flexible with a consequent increase in the dynamic response of the structure. Further analyses are required to evaluate the impact of the material and element nonlinearity on tall buildings subjected to partially correlated wind loads.’